# Merging Generated and Retrieved Knowledge for Open-Domain QA

**Yunxiang Zhang[*], Muhammad Khalifa[*], Lajanugen Logeswaran[†],**
**Moontae Lee[†‡], Honglak Lee[*†], Lu Wang[*]**
University of Michigan[*], LG AI Research[†], University of Illinois at Chicago[‡]

## Abstract

Open-domain question answering (QA) systems are often built with retrieval modules. However, retrieving passages from a given source is known to suffer from insufficient knowledge coverage. Alternatively, prompting large language models (LLMs) to generate contextual passages based on their parametric knowledge has been shown to improve QA performance. Yet, LLMs tend to "hallucinate" content that conflicts with the retrieved knowledge. Based on the intuition that answers supported by both sources are more likely to be correct, we propose COMBO, a ***Compatibility-Oriented knowledge Merging for Better Open-domain QA*** framework, to effectively leverage the two sources of information. Concretely, we match LLM-generated passages with retrieved counterparts into *compatible pairs*, based on discriminators trained with silver compatibility labels. Then a Fusion-in-Decoder-based (Izacard and Grave, 2021b) reader model handles passage pairs to arrive at the final answer. Experiments show that COMBO outperforms competitive baselines on three out of four tested open-domain QA benchmarks. Further analysis reveals that our proposed framework demonstrates greater efficacy in scenarios with a higher degree of knowledge conflicts.[1]

## 1  Introduction

Open-domain question answering (QA) typically requires models to consolidate and reason over information from external knowledge sources (Chen et al., 2017; Petroni et al., 2021). A common *retrieve-then-read* framework (Izacard and Grave, 2021a,b; Izacard et al., 2022; Karpukhin et al., 2020) would first fetch relevant passages from an external corpus and pass them to a reader model to arrive at an answer. Retrieving from reliable

---

[*]Correspondence to yunxiang@umich.edu
[1]Our code is publicly available at https://github.com/yunx-z/COMBO.

Figure 1: A Fusion-in-Decoder-based (Izacard and Grave, 2021b) QA model is misled by the *knowledge conflict* between retrieved passage and hallucinating LLM-generated passage on an example from NaturalQuestions (Kwiatkowski et al., 2019). This is because QA models tend to favor LLM passages when conflict exists. More detailed analyses are in Appendix A.1.

sources, e.g., Wikipedia, enjoys the benefits of being factual, but may suffer from incomplete knowledge coverage and contain irrelevant information.

Large Language Models (LLMs) have been shown to store a wide range of knowledge in their parameters, which can serve as an alternative information source (Brown et al., 2020; Chowdhery et al., 2022; OpenAI, 2023; Ouyang et al., 2022). Capitalizing on the success of leveraging LLMs' parametric knowledge for natural language understanding tasks, a new paradigm known as *generate-then-read* (Yu et al., 2023) has emerged. It prompts an LLM to generate contextual passages for a question in lieu of retrieval from static corpora. Compared to retrieved passages, LLM-generated texts are more relevant to the question, as its generative nature implicitly optimizes for content relevance. Yet, they frequently contain factual errors due to hallucinations (Ji et al., 2023; Peng et al., 2023).

This work aims to address *how to combine retrieved and parametric knowledge to get the best of both worlds for open-domain QA*. Specifically,

we want to maintain the *factuality* of retrieved knowledge while leveraging the *relevance* of LLM knowledge. Yu et al. (2023) have explored a simple merging approach by separately encoding the two types of passages into a Fusion-in-Decoder (Izacard and Grave, 2021b) reader (Figure 2a). While this setting outperforms using either source alone, the direct merging approach ignores inconsistent facts between the sources, which can easily confuse the reader model. Figure 1 shows an example of *knowledge conflicts* where LLM-generated passage contains fabrication, contradicting the retrieved knowledge.[2]

To address this challenge, we propose a novel **C**ompatibility-**O**riented knowledge **M**erging for **B**etter **O**pen-domain (COMBO) QA framework. Intuitively, an answer tends to be correct if it is supported by information from both sources. Therefore, we promote the model's usage of factual and relevant information by combining retrieved and generated knowledge into *compatible pairs*, which are then fed into a reader module.

To date, there has been no gold-standard annotation for training a compatibility scorer for passage pairs. To this end, we introduce a novel method that automatically mines *silver labels* of compatibility at scale, without the need for human annotation or dataset-specific heuristics. Specifically, we estimate the silver compatibility by checking whether the prediction correctness of a QA model would flip if one or both passages from a target pair were to be removed from the input. Afterward, we train discriminators on silver labels to compute passage pairs' compatibility scores.

Lastly, we benchmark COMBO in a fully-supervised setting on four popular open-domain QA datasets, including both single-hop (NaturalQuestions (Kwiatkowski et al., 2019), TriviaQA (Joshi et al., 2017), WebQuestion (Berant et al., 2013)) and multi-hop questions (HotpotQA (Yang et al., 2018)). Using state-of-the-art retrievers, DPR (Karpukhin et al., 2020) and MDR (Xiong et al., 2021), as well as performant LLMs, e.g., InstructGPT (Ouyang et al., 2022) and ChatGPT, COMBO outperforms competitive baselines on all testbeds except HotpotQA by up to +1.9 exact match score.

To summarize, the main contributions of our work are three-fold:

- We introduce COMBO, a novel compatibility-oriented framework for merging LLM knowledge and retrieved knowledge in open-domain QA.

- We automatically mine silver labels for training compatibility scorers without human annotations.

- We demonstrate the effectiveness of our framework with extensive experiments and analyses over four open-domain QA datasets.

## 2 Related Work

**Parametric and Retrieved Knowledge for QA.** QA models commonly have access to two knowledge sources: parametric knowledge stored in language models and retrieved knowledge from external corpora. Previous work (Chen et al., 2022; Longpre et al., 2021; Pan et al., 2021) focuses on *analyzing* knowledge conflicts in simulated settings by perturbing retrieved contexts, e.g., replacing entities with incorrect counterparts. These studies reveal that reader models overly rely on the parametric knowledge and demonstrate limited abilities of resolving conflicting information. In contrast, we propose solutions to repress models from using generated knowledge that contradicts with retrieved contexts. Other works (Li et al., 2022a; Mallen et al., 2022; Neeman et al., 2022) leverage predefined rules to guide the models to choose between parametric vs. retrieved knowledge sources under conflicts. Instead of relying on dataset-specific heuristics, we introduce a generalizable approach that teaches QA models to prioritize compatible information over conflicting ones.

**Augmentation with Large Language Model Outputs.** LLMs, such as InstructGPT (Ouyang et al., 2022) or PaLM (Chowdhery et al., 2022), store rich world knowledge in their model weights and demonstrate impressive performance on open-domain QA tasks with prompting techniques. Recent studies further boost their performance by eliciting supporting evidence before answer prediction. The types of supporting evidence include encyclopedia knowledge (Li et al., 2022b; Sun et al., 2022; Yu et al., 2023), commonsense knowledge (Liu et al., 2022a,b; Shwartz et al., 2020; Wang et al., 2022; West et al., 2022) and chain-of-thought reasoning steps (Magister et al., 2022; Trivedi et al., 2022; Wei et al., 2022; Zhang et al., 2022). Most

---

[2]Figure 8 in Appendix A shows another example of knowledge conflict on HotpotQA (Yang et al., 2018).

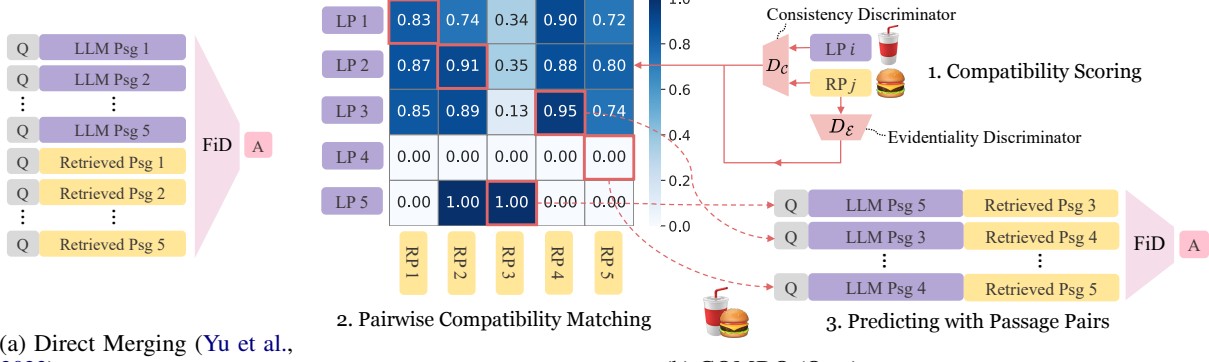

(a) Direct Merging (Yu et al., 2023)

(b) COMBO (Ours)

Figure 2: Overview of COMBO framework (right) that uses both LLM-generated passages and retrieved passages for open-domain QA. 1. We compute pairwise **compatibility** scores, which quantify the extent to which the two passages support each other regarding evidence related to the question. 2. We then match them into pairs by maximizing their overall compatibility (matched pairs are highlighted in red boxes). 3. Finally, passage pairs are sorted by their compatibility and fed to a FiD-based (Izacard and Grave, 2021b) reader model to produce an answer.

relevant to our work, Yu et al. (2023) directly augment retrieval passages with LLM-generated texts, but they do not take knowledge conflicts into account, which could mislead the model and degrade the prediction. Our work promotes the model's usage of factual evidence over hallucinating information in LLM outputs (Creswell and Shanahan, 2022; OpenAI, 2023; Peng et al., 2023; Zhao et al., 2023) with the help of retrieval knowledge.

**Evidentiality-guided QA.** Traditionally, the field of Open-Domain QA has been dominated by *retrieve-then-read* models (Izacard et al., 2021; Izacard and Grave, 2021a; Karpukhin et al., 2020; Xiong et al., 2021). The performance of the readers largely depends on the *evidentiality* of the retrieved passages—whether they contain the correct evidence to support the answer. Recent work (Asai et al., 2022; Fajcik et al., 2021; Lee et al., 2021) augments QA performance by adding a (silver) evidentiality signal of each retrieved passage to the training of reader models. However, their method is insufficient to work with LLM-generated passages that contain hallucination errors. We further incorporate the evidentiality of LLM-generated passages and leverage both sources to highlight the correct evidence for the reader.

## 3 Method

In this section, we present COMBO, which combines both the retrieved passages and the LLM's parametric knowledge to improve open-domain QA performance over the existing retrieve-then-

read and generate-then-read counterparts.[3] The core idea is to feed paired passages—one from an LLM, one from a retriever—into a FiD-based reader model. The passages are paired in such a way that they provide consistent evidence to the question for the reader to identify both the factual and relevant information. Specifically, for each question, $N$ retrieved passages and $M$ LLM-generated passages are given to COMBO. The answer is produced in three steps shown in Figure 2b.

1. The compatibility scores of all possible passage pairs are computed according to an *evidentiality discriminator* and a *consistency discriminator* (§3.1), trained using silver labeled data (§3.2).

2. A *compatibility-oriented matching* strategy selects passage pairs to maximize overall compatibility while balancing the usage of all passages (§3.3).

3. A FiD-based reader handles passage pairs, sorted by compatibility scores, to generate the final answer.

### 3.1 Defining Compatibility

We first describe two assumptions, based on a preliminary analysis provided in Appendix A.4, which lay the basis for the compatibility formulation.[4]

---

[3]We provide a brief introduction to the retrieved-then-read and generate-then-read QA frameworks in Appendix A.2.

[4]For simplicity, we illustrate our method under the single-hop QA setting in the main paper. Appendix A.3 shows how to adapt it to the multi-hop setting with minimal modifications.

**Assumption 1** *Retrieved passages are faithful to real-world facts, i.e., being factual, regardless of its relevance to the question.*

**Assumption 2** *LLM-generated passages contain relevant evidence to the question, i.e., being plausible, irrespective of its factuality.*

Figure 1 shows an example of Assumption 2, where a plausible answer ("Alexandra Breckenridge") is supported by the LLM-generated text despite its inconsistency with the retrieved passage.

With these two assumptions, when an answer can be evinced by *both* the retrieved passage and the LLM-generated passage, it is likely that the retrieved knowledge contains the relevant evidence whereas the generated knowledge is factual. Moreover, the answer tends to be correct in such situations. Therefore, we define the compatibility of a pair of passages as follows.

**Definition 1** *A passage pair is* **COMPATIBLE** *if both passages contain the proper evidence to support answering the question correctly.*

With this concept, we hope to match two passages, one from each source, into compatible pairs. The pairs will facilitate a reader model by promoting the correct evidence when knowledge conflicts exist.

**Compatibility Formulation.** To illustrate how we compute passage compatibility, we formulate the concept of compatibility with mathematical notations. Given a question $Q$, we use $lp_i$ to denote the $i$-th LLM-generated passage for $Q$, and $rp_j$ for the $j$-th retrieved passage. $lp_i \vDash Q$ means $lp_i$ contains the correct evidence for $Q$, and similarly for $rp_j \vDash Q$. Following Definition 1, given $Q$, for a pair of LLM-generated and retrieved passages $lp_i$ and $rp_j$, we formulate the **compatibility score** $c_{i,j}^Q$ as $P(lp_i \vDash Q, rp_j \vDash Q)$, which could be further factorized into two components:[5]

$$
\overbrace{P(lp_i \vDash Q, rp_j \vDash Q)}^{\text{compatibility score}} = \\
\underbrace{P(rp_j \vDash Q)}_{\text{evidentiality score}} \cdot \underbrace{P(lp_i \vDash Q \mid rp_j \vDash Q)}_{\text{consistency score}}.
\tag{1}
$$

$P(rp_j \vDash Q)$ measures whether the retrieved passage contains the correct evidence (i.e., **evidential-**

[5]Notice that we do not factorize the compatibility score the other way round (i.e., $P(lp_i \vDash Q) \cdot P(rp_j \vDash Q \mid lp_i \vDash Q)$). This is because $lp_i$ always contains plausible evidence to the question (Assumption 2) so the discriminator cannot measure the evidentiality of $lp_i$ on its own ($P(lp_i \vDash Q)$). Instead, we use a retrieved factual passage (Assumption 1) to determine whether the $lp_i$ is hallucinating or not.

---

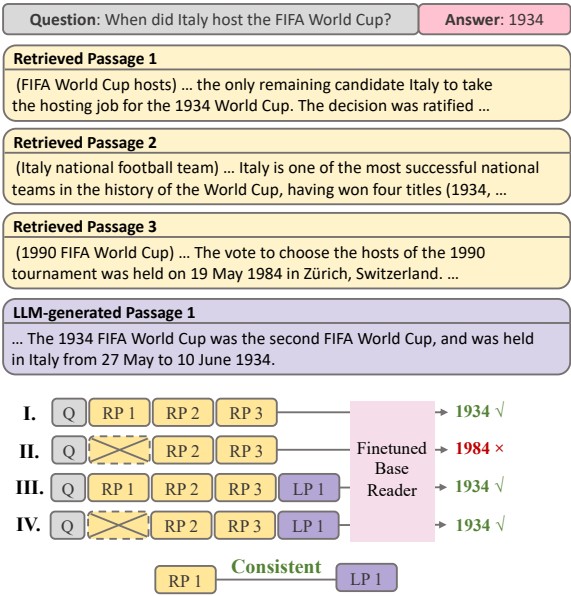

Figure 3: Example for demonstrating the construction of silver consistent labels to train the consistency discriminator $\mathcal{D}_{\mathcal{C}}$.

**ity).** $P(lp_i \vDash Q \mid rp_j \vDash Q)$ characterizes the **consistency** between the retrieved and LLM-generated knowledge.

Based on the decomposition in Equation 1, we train two binary discriminators accordingly: an **evidentiality discriminator** $\mathcal{D}_{\mathcal{E}}$ for modeling $P(rp_j \vDash Q)$ and a **consistency discriminator** $\mathcal{D}_{\mathcal{C}}$ for modeling $P(lp_i \vDash Q \mid rp_j \vDash Q)$. Concretely, $\mathcal{D}_{\mathcal{E}}$ takes a question $Q$ and the $j$-th retrieved passage $rp_j$ as input, and predicts whether $rp_j$ is evidential (positive) or non-evidential (negative). If $rp_j$ is evidential, $\mathcal{D}_{\mathcal{C}}$ will take $Q$, the $i$-th LLM-generated passage $lp_i$, and $rp_j$ as input, and predicts whether $(lp_i, rp_j)$ is consistent (positive) or conflicting (negative), i.e., the retrieved passage contains the correct evidence while the LLM-generated passage does not. As a result, a passage pair is considered compatible if it is both evidential and consistent.

## 3.2 Mining Silver Labels

We do not have human-annotated evidentiality and consistency labels in most existing datasets. To avoid costly manual annotation, we propose an approach to automatically mine silver labels for training the two discriminators. To collect silver evidentiality labels for training $\mathcal{D}_{\mathcal{E}}$, we largely follow the leave-one-out generation approach by Asai et al. (2022). Briefly, a passage is deemed **EVIDENTIAL** if the prediction by a given QA model

changes from correct to incorrect after removing it from a pool of retrieved passages. We explain the details in Appendix C.1.

We then extend this idea to mine consistency labels of passage pairs for training $\mathcal{D}_C$, as demonstrated in Figures 3 and 9 (for negative sample, in Appendix C.2). Intuitively, if the prediction changes from correct to incorrect after removing either passage from the target pair (e.g., IV⇒II for an LLM-generated passage and I⇒II for a retrieved passage, as in Figure 3), both passages supposedly contain the correct evidence and they are therefore a **CONSISTENT** pair. Concretely, for each dataset, we first finetune a base reader model initialized from Fusion-in-Decoder (Izacard and Grave, 2021b) with top-$N$ retrieved passages $\mathbf{P}_R = \{rp_1, rp_2, ..., rp_N\}$ as input.[6] We then label the consistency of $(lp_i, rp_j)$ based on the correctness of predictions with four types of inputs: I. all the retrieved passages $(Q, \mathbf{P}_R)$ II. all the retrieved passages except the target retrieved passage $(Q, \mathbf{P}_R \backslash \{rp_j\})$ III. all the retrieved passages plus the target LLM-generated passage $(Q, \mathbf{P}_R \cup \{lp_i\})$ IV. all the retrieved passages without the target retrieved passage and with the target LLM-generated passage $(Q, \mathbf{P}_R \cup \{lp_i\} \backslash \{rp_j\})$. We consider $(lp_i, rp_j)$ as consistent if the model's prediction is incorrect with II and all correct with I, III, and IV. Conversely, $(lp_i, rp_j)$ is labeled as **CONFLICTING** if the prediction is correct with I and all incorrect with II, III, IV.[7]

### 3.3 Matching Retrieved and LLM-generated Passages into Pairs

To signal the reader model with the connections between the two knowledge sources (compatible vs. conflicting), we adapt FiD to encode passage pairs as input. Each pair of passages is concatenated with the question, and processed independently from other pairs by the encoder. We add special tokens: "question:", "generated passage:" and "retrieved passage:" before the question and the passages, respectively. In this way, the encoder could perform self-attention (Vaswani et al., 2017) across the retrieved and LLM-generated passages. We keep the relative order of LLM-generated and retrieved passage fixed across all input pairs, by always putting $lp_i$ in front of $rp_j$ (Figure 2b). This

trains the model to rely on the factual evidence that is always located in the latter passage when facing conflicting information.

To create passage pairs with a balance of factuality and coverage of supporting evidence, we design a **compatibility-guided optimal matching** strategy with the following two goals. First, for simplicity, we include all passages during the matching process, leaving it to future work to filter incompatible passage pairs. Second, we aim to maximize the compatibility of matched passage pairs, to minimize the adverse impact of knowledge conflicts on the reader model.

To do so, we first use the two discriminators together to compute scores for evaluating the compatibility of all possible pairwise combinations of the retrieved and LLM-generated passages of a question (i.e., $\{(lp_i, rp_j) \mid i \in \{1, 2, ..., M\}, j \in \{1, 2, ..., N\}\}$). This results in a 2-dimensional matrix shown in Figure 2b, where each element represents the compatibility score for a passage pair. A subset of all possible $M \times N$ pairs (red boxes highlighted in the matrix of Figure 2b) are then selected as input to the reader model. The selection is solved as a bipartite graph maximum-weighted matching problem, with details in below paragraph. The algorithm requires that compatible pairs score higher than conflicting and non-evidential ones, so it can achieve the maximum overall compatibility of matched passage pairs while balancing the usage of all passages. To this end, we devise a simple heuristic for compatibility scoring, dubbed as *evidentiality-cutoff*:

$$c_{i,j}^Q = P(lp_i \vDash Q \mid rp_j \vDash Q) \cdot \mathbb{1}_{\{P(rp_j \vDash Q) > 0.5\}} \tag{2}$$

where $\mathbb{1}$ is an indicator function. It binarizes the decision from $\mathcal{D}_\mathcal{E}$ to score non-evidential pairs as zero and prioritize compatible pairs over conflicting counterparts.

**Compatibility-guided Optimal Matching**    We view the compatibility-guided matching process as a maximum-weighted matching problem on a bipartite graph. Concretely, we treat each passage as a node and there are edges connecting LLM-passage-nodes with retrieved-passage-nodes, whose weights are the corresponding compatibility score. This results in a complete bipartite graph $G = (\mathbf{P}_L, \mathbf{P}_R; E)$ where $E = \{(lp_i, rp_j) \mid i \in \{1, 2, ..., M\}, j \in \{1, 2, ..., N\}\}$. The weight function $w : E \to [0, 1]$ assigns compatibility score of

---

[6]In our experiments, we set $N = 10$ across all datasets.

[7]To speed up the consistency label mining process, we only need to obtain model predictions for III and IV if I is correct and II is incorrect (i.e., $rp_j$ is evidential).

Equation 2 as edge weight: $w((lp_i, rp_j)) = c_{i,j}^Q$. We want to find a perfect matching $\mathcal{M}$ of maximum weight where the weight of matching $\mathcal{M}$ is given by $w(\mathcal{M}) = \sum_{e \in \mathcal{M}} w(e)$. This problem is typically solved by the Hungarian algorithm (Kuhn, 1955, 1956) in polynomial time. This matching is optimal in the sense that it covers all passages while maximizing the sum of their compatibility scores.

## 4 Experiments

### 4.1 Experimental Setup

We experiment with four open-domain QA datasets: **NaturalQuestions** Open (Kwiatkowski et al., 2019), **TriviaQA** unfiltered (Joshi et al., 2017), **WebQuestion** (Berant et al., 2013) for single-hop QA and **HotpotQA** full wiki setting (Yang et al., 2018) for multi-hop QA. We provide brief introductions and statistics for each dataset in Appendix B. We use Exact Match (EM) (Rajpurkar et al., 2016) as our major evaluation metric across all datasets. We run each experiment three times with different random seeds and report the average.[8]

We focus on the fully-supervised setting and obtain retrieved and LLM-generated passages for each question in the dataset. For retrieved passages, we use retrieval results by DPR (Karpukhin et al., 2020) for single-hop QA and MDR (Xiong et al., 2021) for multi-hop QA, where no gold-standard passages are used in this study. We do not further finetune the retriever. For LLM-generated passages, we use the ones provided by Yu et al. (2023)[9] for the three single-hop QA datasets, which are based on InstructGPT. Since there are no available resources for LLM-generated passages on multi-hop QA, we obtain them by calling ChatGPT's API (`gpt-35-turbo`). We choose ChatGPT because it is a performant LLM with reasonable costs.

We employ FiD (Izacard and Grave, 2021b) as our reader model. For the discriminators $\mathcal{D}_\mathcal{E}$ and $\mathcal{D}_\mathcal{C}$, we use RoBERTa-large (Liu et al., 2019) for single-hop QA and DeBERTa-large (He et al., 2021) for multi-hop QA. DeBERTa features relative positional embedding, making it possible to adapt to longer input for passages of multi-hop QA.

In our experiments, we train separate discriminators $\mathcal{D}_\mathcal{E}$ and $\mathcal{D}_\mathcal{C}$ for each dataset. However, given the limited amount of silver labels mined from the small-scale dataset WebQuestions, we warm up

---

[8]For results of standard deviation under EM and F1 scores, please see Table 6 in Appendix D.1.
[9]https://github.com/wyu97/GenRead.git

| Methods | NQ test | TQA test | WebQ test | HQA all $Q$ dev | HQA bridge $Q$ dev |
|---|---|---|---|---|---|
| *Single Knowledge Source* | | | | | |
| Retrieved Psg. Only | 46.7 | 61.9 | 48.1 | 59.9 | 55.4 |
| LLM Psg. Only | 40.3 | 67.8 | 51.5 | 42.6 | 35.9 |
| *Two Knowledge Sources* | | | | | |
| Direct Merging | 52.7 | 74.2 | 51.1 | **61.6** | 57.8 |
| Random Matching | 53.3 | 74.2 | 51.6 | 61.5 | 57.7 |
| COMBO (ours) | **54.2** | **74.6** | **53.0** | **61.6** | **58.0** |

Table 1: Exact Match (EM) results on NaturalQuestions (Kwiatkowski et al., 2019), TriviaQA (Joshi et al., 2017), WebQuestion (Berant et al., 2013), and HotpotQA (Yang et al., 2018). For experiments under *Two Knowledge Sources*, we conduct three runs with different random seeds and report the average.

the training of its discriminators with data from the other two single-hop QA datasets. Note that the additional warm-up data is only used for training the discriminators but not the reader model, so as to ensure a fair comparison with the baselines. More details of the prompts for LLMs, model implementation and hyperparameters are included in Appendix C.2.

### 4.2 Results and Analysis

**Comparison with Baselines.** Table 1 shows experimental results on the four open-domain QA datasets. All methods use 10 retrieved and/or 10 LLM-generated passages for each question $Q$ and FiD-large (770M) as the reader model. Yu et al. (2023) presents a strong baseline of directly merging two knowledge sources. With the improved coverage of world knowledge, it beats the retrieval-only model by an average of 7.1 EM over the three single-hop QA datasets. With the same reader model and the same set of input passages, our COMBO framework further improves over Direct Merging by an average of 1.3 EM scores across the single-hop QA datasets.

We also compare our proposed algorithm to a Random Matching baseline, which randomly divides passages into pairs. Our improvement over Random Matching indicates that compatible pairs of LLM and retrieved passages are essential for model to make better predictions. Appendix D.1 also shows the results of an oracle setting assuming access to ground-truth answers for matching and the state-of-the-art results reported in existing literature. Since our approach does not modify retriever or reader architecture, it can be easily extended to

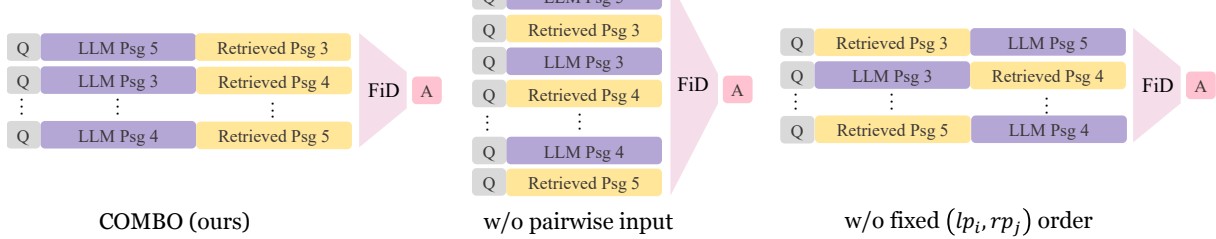

Figure 4: Illustrations of the input formats for two of the ablation experiments in Table 2.

capitalize on larger models, more retrieved passages and additional knowledge sources (e.g., tables and KBs), as already leveraged in state-of-the-art systems. We also note that any potential performance improvements are somewhat constrained by the percentage of the hallucinated (generated) and evidential (retrieved) passages provided. In the extreme scenario where all generated passages are devoid of hallucinations or all retrieved passages lack correct evidence, our method would default to Random Matching, as the compatibility matching score would be uniform.

Different from single-hop QA, we do not observe improvements on HotpotQA. However, minor improvement is observed on the subset (73%) of bridge questions that require the retriever or LLM to find a bridge entity that connects two pieces of evidence together to reveal the answer. It shows that our method is not directly generalizable to the more challenging multi-hop setting, and thus calls for a more fine-grained modeling of the compatibility between hops of evidence. We note that for the rest of comparison questions in HotpotQA (e.g., "*who is younger, Keith Bostic or Jerry Glanville?*"), models could make the right prediction even if the LLM-generated passages contain hallucinated evidence (see Appendix D.2 for an example). This explains why our compatibility-oriented framework appears to be less effective on this subset.[10]

**Ablation Study.** We create several variants to validate the effectiveness of different components of our COMBO framework, by removing or replacing each component individually. Results are shown in Table 2. First, we remove the evidentiality discriminator $\mathcal{D}_\mathcal{E}$ and only use the probability given by the

| Method | NQ dev | TQA dev |
|---|---|---|
| COMBO (ours) | **52.3** | **73.9** |
| w/o evidentiality discriminator | 51.8 | 73.5 |
| w/o pairwise input | 51.5 | 73.4 |
| w/o sorting pairs | 51.7 | 73.7 |
| w/o fixed $(lp_i, rp_j)$ order | 50.8 | 73.3 |
| w/o evidentiality-cutoff | 51.7 | 73.6 |
| w/o optimal matching | 51.6 | 73.6 |

Table 2: Ablation study results on NaturalQuestions (NQ) and TriviaQA (TQA).

consistency discriminator $\mathcal{D}_\mathcal{C}$ as the compatibility score for ranking passage pairs (*w/o evidentiality discriminator*). The performance drop highlights the importance of having an evidentiality discriminator to filter out irrelevant (i.e., non-evidential) retrieved passages before computing the passage consistency and thus echoes our compatibility decomposition motivated by Equation 1.

Second, we remove the pairwise formulation by linearizing the matched pairs into a sequence of single passages as input (*w/o pairwise input*). See Figure 4 in Appendix C.2 for an illustration of the linearized inputs. The performance loss validates the usefulness of the encoder's self-attention between tokens of LLM-generated and retrieved passages, which allows the reader module to jointly reason over passage pairs as aggregated evidence.

Third, we randomly shuffle the order of input passage pairs instead of ranking by their compatibility scores (*w/o sorting pairs*). The score worsens, as this model variant fails to learn to prioritize compatible information. Further analysis of model's behavior in Appendix D.4 verifies that our model could attribute higher attention scores to compatible pairs than others when making predictions.

For the fourth ablation experiment (*w/o fixed $(lp_i, rp_j)$ order*), we randomly shuffle the order of $lp_i$ and $rp_j$ within each passage pair, instead of always putting $lp_i$ in front of $rp_j$ (Figure 4, Ap-

---

[10]We are unable to show results of multiple methods on the hidden test set of HotpotQA, as the official leaderboard of HotpotQA (https://hotpotqa.github.io/) allows submitting predictions only *once*. Therefore, we show results on the dev set instead.

| Conflicting Rate | Subset% | Retrieved Psg. Only | Direct Merging | COMBO |
|---|---|---|---|---|
| 0 − 0.1 | 56.2% | 41.6 | 47.7 | **48.5** (+0.8) |
| 0.1 − 0.2 | 22.7% | 45.1 | 53.1 | **53.3** (+0.2) |
| 0.2 − 0.3 | 15.5% | 52.5 | 59.0 | **59.9** (+0.9) |
| 0.3 − 0.4 | 1.8% | 62.5 | 65.0 | **67.5** (+2.5) |
| 0.4 − 0.5 | 2.2% | 57.4 | 64.0 | **66.0** (+2.0) |
| 0.5 − 1.0 | 1.6% | 61.8 | 56.6 | **61.0** (+4.4) |

Table 3: The performance of our method compared to others on NaturalQuestions dev set w.r.t. conflicting rate (percentage of conflicting passage pairs, Equation 3). Larger improvements of COMBO over Direct Merging are shaded with darker orange. Overall, COMBO's improvement over Direct Merging is greater when the conflicting rate is higher, suggesting the robustness of our method to knowledge conflicts.

pendix C.2). We find that fixing their order allows the model to learn to rely on the factual evidence in the latter retrieved passage when facing conflicting information.

We also explore other ablations to show the contributions of several heuristics employed in our system, including *evidentiality-cutoff* (Equation 2) and *optimal matching* (Section 3.3). We replace Equation 2 with Equation 1 for producing compatibility scores (*w/o evidentiality-cutoff*). Equation 1 directly multiplies the two probabilities given by the two discriminators together. It demonstrates that models benefit from a binarized decision by $\mathcal{D}_{\mathcal{E}}$ and we hypothesize that this is because the raw predicted probability is often not well calibrated (i.e., predicted probabilities do not correlate well with the probabilities of correctness (Guo et al., 2017; Jiang et al., 2021)).

Additionally, we replace the optimal matching (maximum weighted matching) with a greedy strategy (*w/o optimal matching*). Specifically, we first match compatible passage pairs, followed by conflicting and non-evidential pairs. When finding the $k$-th pair to match, we only consider passages not appearing in previous $k-1$ pairs. The performance drop shows the optimal matching could bring together more compatible pairs and thus better guide the model's predictions.

**Impact of Conflicting Contexts on the Reader Model.** Here we aim to answer a question: *How well can the reader model (i.e., FiD) pinpoint the correct answer when varying degrees of knowledge conflicts exist between LLM-generated texts and retrieved passages?* Table 3 shows the performance of different methods when facing different rates of

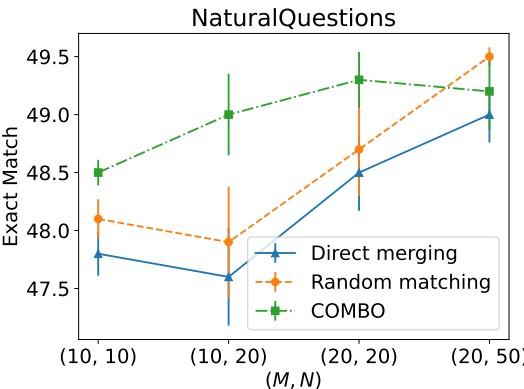

Figure 5: The performance on NaturalQuestions dev set when scaling the number of LLM-generated passages ($M$) and retrieved passages ($N$) for each question. We report the average number and standard deviation (error bar) over three runs with different random seeds.

conflicting knowledge. We measure **conflict rate** as follows:

$$\text{conflicting\_rate} = \frac{N_A \cdot (M - M_A)}{N \cdot M}. \quad (3)$$

$N_A$ refers to the number of retrieved passages that contain the gold answer string $A$ and $M - M_A$ means the number of LLM-generated passages that do not contain the gold answer string. The conflicting rate indicates the percentage of conflicting pairs (i.e., $rp_i$ contains the answer while $lp_j$ does not) over all possible pairs. Table 3 suggests that when there is minimal conflicts between the two sources, both Direct Merging and COMBO can significantly improve over the Retrieved-passage-only model. However, when the conflict rate is high, only COMBO can maintain a consistent improvement over the Retrieved-passage-only baseline, suggesting its robustness.

**Scaling with Number of Passages.** We further evaluate the performance of COMBO with respect to different numbers of LLM-generated passages ($M$) and retrieved passages ($N$). Figure 5 shows the results for NaturalQuestions. Given a larger number of passages, we switch our reader model from FiD-large to FiD-base due to GPU memory limits. When $M$ is smaller than $N$, we simply duplicate the LLM-generated passages to match the number of retrieved passages, so that every passage is included in the matching results (consistent with Section 3.3). We observe that given more passages from both sources, our framework generally achieves greater performance gain over

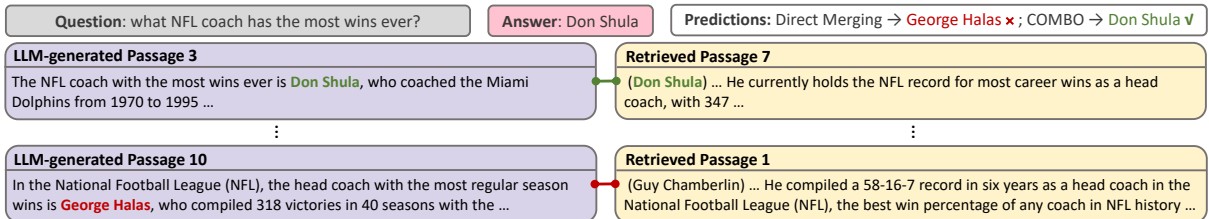

Figure 6: An example of a QA pair and the passage matching results by COMBO. Passage pairs are sorted by their compatibility scores. It shows how COMBO rectifies the prediction of the baseline method under knowledge conflicts by prioritizing compatible pairs (green connecting line) over incompatible pairs (red connecting line).

the direct merging approach. Additional analysis in Appendix D.3 shows that more knowledge conflicts arise from the increased number of input passages, again highlighting the importance of compatibility-guided knowledge merging. Because we train our discriminators only on the top 10 passages, they may not generalize well when applied to the top 50 passages. This is possibly why our method seems less effective when provided with 50 retrieved passages as input.

**Case Study of Matching Results.** Figure 6 shows an example from NaturalQuestions where the COMBO matching results of passages help rectify the prediction of the Direct Merging model. In the first *compatible* pair, both passages contain correct evidence that supports the ground-truth answer "Don Shula". The last one is an *incompatible* pair where the LLM-generated passages frequently contain a hallucinating error ("George Halas") that misleads the Direct Merging model to make a wrong prediction. Since we prioritize compatible pairs over incompatible pairs, the reader model could leverage this inductive bias by paying more attention to the higher-ranked passages.

**Human Evaluation of Compatibility Labels.** Figure 7 shows the confusion matrix of the compatibly labels predicted by our discriminators on 150 examples from the dev set of NaturalQuestions. Specifically, we randomly select 25 questions and sample 2 compatible pairs, 2 conflicting pairs and 2 non-evidential pairs (if applicable) for each question. The authors manually analyze whether the passages actually contain the correct evidence. We find that our discriminators yield an overall accuracy of 78% on this 3-way classification task. It shows that our discriminators are capable of providing signals of passage compatibility to the reader model for making well-informed decisions.

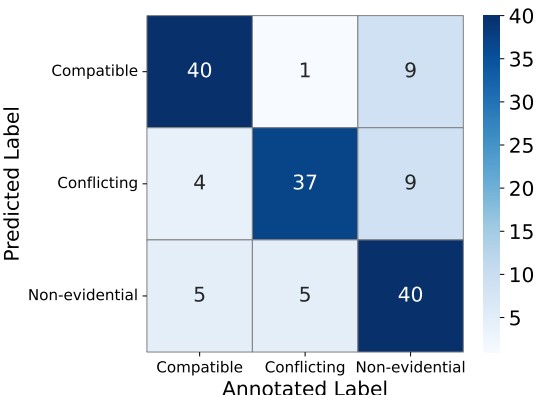

Figure 7: Confusion matrix of the human annotations vs. predicted labels by our discriminators on 150 random samples from NaturalQuestions dev set. The overall accuracy is 78%.

## 5 Conclusion and Future Work

In this work, we study the problem of merging retrieved and LLM-generated knowledge for open-domain QA. We tackle the challenge of knowledge conflicts caused by LLM's hallucination with a compatibility-oriented knowledge merging framework (COMBO). Specifically, we match LLM-generated and retrieved passages for a given question into pairs based on their compatibility and perform information fusion on the encoder side of the FiD-based reader by feeding matched pairs as input. Experiments on four open-domain QA datasets demonstrate the empirical success of COMBO.

In the future, we plan to extend our framework to few-shot QA settings by employing an LLM with in-context learning to compute the compatibility scores instead of training a discriminator from scratch, thereby eliminating the necessity for (weakly) supervised training. But it would also require carefully examining whether and which LLM is able to discern the knowledge conflicts between its own parametric memory and retrieved evidence.

## Limitations

1. We only evaluate our knowledge merging framework on the tasks of open-domain QA. It would be interesting to apply our method to other knowledge-intensive NLP tasks such as fact checking (Thorne et al., 2018) and knowledge-enhanced text generation (Dinan et al., 2019; Yu et al., 2022).

2. Under the conditions of our experiment, we did not manipulate the distribution of questions or input passages in any way. However, we anticipate that in more controlled settings, such as PopQA (Mallen et al., 2023) where questions about infrequent entities are examined in more detail, our method could yield more dramatic performance improvements. At present, LLMs appear more susceptible to hallucinating knowledge pertaining to less frequent entities. We eagerly anticipate investigating this aspect further in our future work.

3. For each dataset, we only conduct experiments with generated passages from a single LLM (either InstructGPT or ChatGPT) and retrieved passages from a single retriever (either DPR or MDR). Our experiments are limited to a generative reader model which is based on the widely-used Fusion-in-Decoder (Izacard and Grave, 2021b) architecture.

4. Regarding computational overheads of COMBO, we leverage the same set of input passages and same reader model as direct merging. During the training of the FiD-based reader model, according to our empirical observations on NaturalQuestions, it only introduces minimal computational overheads of approximately 23% more GPU memory and 27% more training time, compared to direct merging. Practitioners who can afford to train the previous direct merging model should also be able to train our framework.

5. Our proposed approach for silver label mining could result in a limited amount of labels on a small dataset, which could not provide sufficient data to train a dataset-specific discriminator. It thus calls for the need for training a unified discriminator that works for various datasets and tasks.

## Ethics Statement

Large language models are known to have racial and gender biases and may generate harmful content such as fabricated facts and toxic response (OpenAI, 2023). Since we leverage generated contextual passages from LLMs to enhance QA performance, our models may inherit these biases during generation. Although we aim to promote the usage of factual information by combining generated knowledge with retrieved knowledge from an external trustworthy corpus, we do not fully eliminate hallucinations from LLMs. Interested practitioners should carefully address these risks before deploying our framework in certain domains such as politics, finance, and healthcare.

## Acknowledgements

This work is supported by LG AI Research and computational resources and services provided by Advanced Research Computing (ARC), a division of Information and Technology Services (ITS) at the University of Michigan, Ann Arbor. Additionally, we would like to thank Wenhao Yu for his invaluable assistance in clarifying aspects of their published work and generously sharing code and data to facilitate the reproducibility of the results presented in this paper. We also appreciate the anonymous reviewers for their valuable suggestions. We thank the members of the LAUNCH group at the University of Michigan for their discussions and suggestions.

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

## A Preliminaries

### A.1 Analysis of Reader's Predictions under Knowledge Conflicts

We are interested in the question: *does FiD-based reader model prefer LLM-generated passages over retrieved ones for making predictions under knowledge conflicts?* We compare the outputs of two models: one is fed with retrieved passages only, and the other takes the concatenation of retrieved passage and generated passage as input (i.e., Direct Merging shown in Figure 2a). We focus on the examples from dev set of NaturalQuestions where knowledge conflicts probably exist. Here we select questions whose retrieved passages contain correct answer while LLM-generated passages do not. Specifically, we obtain a subset of questions with more then 20% of conflicting passage pairs (i.e., conflicting rate defined in Equation 3). On this subset, the Retrieved-passage-only model is not affected by knowledge conflicts since it does not have access to LLM-generated passages.

We find that on the examples initially predicted as correct by the Retrieved-passage-only method, models are fooled by adding LLM-generated passages on 16% of them (i.e., Direct Merging method gives the wrong predictions). The performance drop shows that there is still room for improvement for FiD-based reader to resolve conflicts on its own, thus calling for the need of our compatibility-oriented QA framework.

Additionally, we observe that 65% of predictions are sub-spans of at least one of the generated passages. This shows that the reader favors LLM passages more than retrieved passages when conflict exists. LLM passages often appear to be more relevant to the question and describe the plausible evidence in a way more specific to question. We hypothesize that the reader model find it easier to extract answer from LLM-generated passages so it tend to rely on them more for predictions.

### A.2 Retrieve/Generate-then-read Open-domain QA Framework

The task of open-domain QA is typically solved by a retrieve-then-read framework (Karpukhin et al., 2020) consisting of two modules: 1) a retriever $R$ that fetches top-$N$ relevant passages $\mathbf{P}_R = \{rp_1, rp_2, ..., rp_N\}$ to the question $Q$ from a large corpus $\Omega_R$ such as Wikipedia and 2) a generative reader $\mathcal{G}$ that generates the answer $A$ conditioned on retrieved passages $\mathbf{P}_R$, denoted

**HotpotQA (Multi-hop QA)**

**Question**: Gail Matthius co-anchored the Weekend Update segment of "Saturday Night Live" with the actor who played the villain Nicholas Andre in what movie?
**Answer**: Dumb and Dumber ✓
**Prediction**: The Goonies ✗

**Retrieved Passage**

1. (Gail Matthius) ... co-anchored the Weekend Update segment with Charles Rocket in 1981.
2. (Charles Rocket) ... He was best known for his role as the villain Nicholas Andre in the film "**Dumb and Dumber**" ...

**LLM-generated Passage**

1. Gail Matthius is an American actress ... She also co-anchored the Weekend Update segment of the show "Saturday Night Live" ...
2. Robert Davi ... is perhaps best known for his roles in movies such as ... "**The Goonies**" ... where he played the character Nicholas Andre.

Figure 8: A Fusion-in-Decoder-based (Izacard and Grave, 2021b) QA model is misled by the *knowledge conflict* between retrieved passage and hallucinating LLM-generated passage on an example from HotpotQA (Yang et al., 2018).

as $A = \mathcal{G}(Q, \mathbf{P}_R)$. We use Fusion-in-Decoder (FiD) (Izacard and Grave, 2021b), a state-of-the-art retrieval-augmented generation model, as our reader model. FiD is a sequence-to-sequence architecture based on the pretrained T5 models (Raffel et al., 2020). It first encodes separately each passage $r_i$ appended to the question, resulting in representation $\mathbf{p}_j = \text{T5-encoder}(Q, rp_j)$. Then the decoder attends to the concatenation of representations of all passages and generates the answer: $A = \text{T5-decoder}(\mathbf{p}_1, \mathbf{p}_2, ..., \mathbf{p}_N)$. The cross-attention mechanism of the decoder enables FiD to perform evidence aggregation, i.e., assigning different attention scores to each passage.

The recently proposed generate-then-read pipeline (Yu et al., 2023) simply substitutes the retriever $R$ with an LLM $L$ that generates top-$M$ contextual passages $\mathbf{P}_L = \{lp_1, lp_2, ..., lp_M\}$ expressing knowledge stored in its parameters. Then these LLM-generated passages are handled by the reader to give the final answer: $A = \mathcal{G}(Q, \mathbf{P}_L)$. Yu et al. (2023) also explore a simple approach (Figure 2a) of directly merging passages from both sources together: $A = \mathcal{G}(Q, \mathbf{P}_L \cup \mathbf{P}_R)$, regardless of knowledge conflicts between $\mathbf{P}_L$ and $\mathbf{P}_R$.

### A.3 Extension of Compatibility Definition from Single-hop to Multi-hop QA

Different from the single-hop QA setting, our retrieved item under the multi-hop QA setting is actually a passage chain $rc_j = (rp_j^1, rp_j^2, ..., rp_j^k)$ instead of a single passage $rp_j$, following the common practice proposed by Xiong et al. (2021).

| Settings | Single-hop QA | Multi-hop QA |
|---|---|---|
| Assumptions | • Retrieved passages are factual.
• LLM-generated passages contain relevant evidence to the question. | • Retrieved passage chains are factual.
• LLM-generated passage chains contain a relevant reasoning path to the question. |
| Definitions | Given a question $Q$, LLM-generated passage $lp_i$ and retrieved passage $rp_j$ are ...
• COMPATIBLE if both $lp_i$ and $rp_j$ contain the correct evidence to support ground-truth answers of the question $Q$
• CONFLICTING if $rp_j$ contains correct evidence while $lp_i$ contains incorrect evidence
• NON-EVIDENTIAL if $rp_j$ does not contain correct evidence | Given a question $Q$, LLM-generated passage chain $lc_i$ and retrieved passage chain $rc_j$ are ...
• COMPATIBLE if both $lc_i$ and $rc_j$ contain the correct reasoning path to support the ground-truth answers of the question $Q$
• CONFLICTING if $rc_j$ contains correct reasoning path while $lc_i$ contains incorrect reasoning path
• NON-EVIDENTIAL if $rc_j$ does not contain correct reasoning path |

Table 4: Comparision of definitions and assumptions for compatibility between single-hop and multi-hop QA settings.

Specifically, in HotpotQA (Yang et al., 2018), all questions are supposed to be 2-hop, so a passage chain $rc_j$ is essentially an ordered pair of passages $(rp_j^1, rp_j^2)$ that provides sufficient evidence for answering $Q$. An example of the retrieved passage chain is shown in Figure 8. We modify the assumptions and definitions for the multi-hop setting accordingly, as shown in Table 4. Specifically, one could simply replace $rp_j$ with $rc_j$ and $lp_i$ with $lc_i$ throughout this paper to obtain the same conclusion. The modifications of definitions from the single-hop to the multi-hop setting are minimal yet reasonable, allowing for a unified concept formulation and system implementation to handle both single-hop and multi-hop QA.

### A.4 Validating Assumptions behind the Compatibility Definition

In Section 3.1, we make two assumptions for computing passage compatibility. Assumption 1 ("*retrieved passages are factual*") is widely adopted in the existing literature (Ji et al., 2023; Lee et al., 2022; Thorne et al., 2018) when retrieving from Wikipedia corpus. For Assumption 2 ("*LLM-generated passages contain relevant evidence to the question*"), the authors manually annotate 100 random examples from the NaturalQuestions (Kwiatkowski et al., 2019) dataset to verify the plausibility of LLM's generations. For each example, we first determine whether it contains a plausible answer. If yes, we further compare it to the ground-truth answer or search on Wikipedia to label it as hallucinating or not. We show examples of annotations in Table 5. We find that 94% of them contain plausible answers, thus supporting our assumption. Of all the passages that contain plausible answers, 51% of the answers are

incorrect, meaning that these passages suffer from the hallucinations of LLM and pose a risk to the reliability of QA readers. This highlights the challenge of knowledge merging and the importance of our compatibility-oriented matching framework COMBO.

## B Dataset Details

We use three single-hop QA datasets (NaturalQuestions (Kwiatkowski et al., 2019), TriviaQA (Joshi et al., 2017), WebQuestions (Berant et al., 2013)) and one multi-hop (HotpotQA (Yang et al., 2018)) QA dataset as the testbeds for evaluating our method. NaturalQuestions consists of real-world information-seeking queries issued to the Google search engine and their corresponding long answers (gold evidence passage) and short answers (one or more entities). We use the open-domain version created by Lee et al. (2019) which only keeps questions and short answers with less than five tokens. TriviaQA includes questions from trivia and quiz-league websites. WebQuestions contains questions from Google Suggest API that queries entities in Freebase. HotpotQA contains questions requiring reasoning over multiple Wikipedia documents to answer. We focus on the fullwiki setting where QA systems need to retrieve relevant evidence from the whole Wikipedia corpus. For train / dev / test splits, we use the same setting as Karpukhin et al. (2020) for NaturalQuestions, TriviaQA, and WebQuestions, and the official leaderboard version[11] for HotpotQA.

---

[11] https://hotpotqa.github.io/

| Type | Question | Gold Answer | LLM-generated Passage | Plausible Answer | Plau. | Hallu. | Perc. |
|------|----------|-------------|----------------------|------------------|-------|--------|-------|
| I | Who played Floyd on the Andy Griffith show? | Howard Terbell McNear | Floyd Lawson was a recurring character on The Andy Griffith Show. He was played by Walter Brennen. | Walter Brennen | Y | Y | 48% |
| II | When did the new 3DS XL come out? | February 13, 2015 | The New Nintendo 3DS XL was released on February 13, 2015, in Japan and North America. | February 13, 2015 | Y | N | 46% |
| III | Who shifted the capital from Calcutta to Delhi? | Government of British India | In 1911, the British imperial capital was shifted from Calcutta to Delhi. This was done as a strategic move to make administration more efficient. The move was also seen as an effort to co-opt and acculturate the growing Indian middle class into the colonial project. | - | N | - | 6% |

Table 5: Examples of annotations for plausibility (Plau.) and hallucinations (Hallu.) of LLM-generated passages. Analysis is performed on 100 random samples from the dev set of NaturalQuestion.

## C   Implementation Details

### C.1   Silver Evidentiality Label Mining

We use the same reader $\mathcal{G}_b$ for consistency label mining in Section 3.2. Inspired by Asai et al. (2022), we consider $rp_j$ as evidential for $Q$ if 1) the prediction based on top-$N$ retrieved passages $\mathcal{G}_b(Q, \mathbf{P}_R)$ is correct and 2) $\mathcal{G}_b(Q, \mathbf{P}_R\backslash\{rp_j\})$, the prediction based top-$N$ retrieved passages except $rp_j$, is incorrect. $rp_j$ is thus believed to contain the correct evidence since removing itself makes the predictions change from right to wrong. Similarly, we consider $rp_j$ as non-evidential for $Q$ if $\mathcal{G}_b(Q, \mathbf{P}_R)$ is incorrect and $\mathcal{G}_b(Q, \mathbf{P}_R\backslash\{rp_j\})$ is correct, which indicates adding $rp_j$ to the rest of input passages misleads the model.

### C.2   Model Implementations and Hyperparameters

For LLM-generated passages, we use the ones provided by Yu et al. (2023)[12] for the three single-hop QA datasets. They prompt InstructGPT (`text-davinci-002`) (Ouyang et al., 2022) with the prompt "`Provide a background document from Wikipedia to answer the given question. \n\n {question} \n\n`" and use sampling to generate 20 documents for each question. Additionally, we prompt ChatGPT (`gpt-3.5-turbo`) to generate contextual passages for HotpotQA. We use the prompt "`You are an assistant designed to provide a chain of two 100-word documents from Wikipedia`

[12]https://github.com/wyu97/GenRead.git

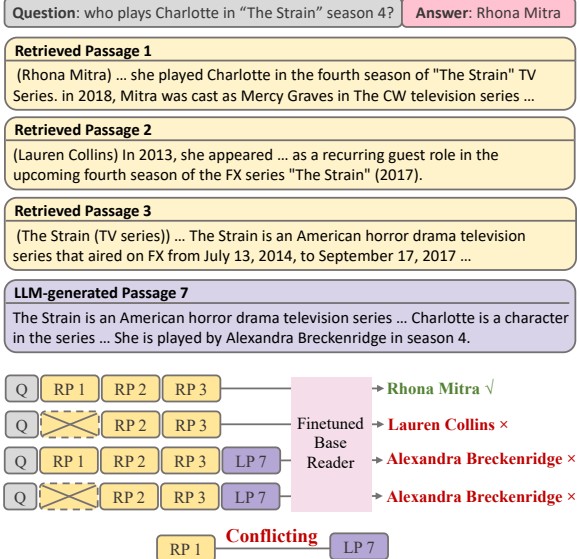

Figure 9: Overview and examples of our proposed approach for mining silver conflicting labels. For a pair of target retrieved and LLM-generated passages (e.g., RP 1 and LP 7 as in this Figure), we create four types of input by I. using all the retrieved passages, II. masking the target retrieved passage, III. appending the target LLM-generated passage, IV. doing both II and III. We obtain predictions by a finetuned reader model when using different types of input. If the prediction is correct with I but all incorrect with II, II, IV, we label it as a conflicting pair. See Figure 3 in Section 3.2 for an example of consistent pair.

| Methods | NQ test | TQA test | WebQ test | HQA all $Q$ dev | HQA bridge $Q$ dev |
|---|---|---|---|---|---|
| Direct Merging | $52.7_{0.60}$ / $61.2_{0.33}$ | $74.2_{0.10}$ / $80.5_{0.13}$ | $51.1_{0.30}$ / $58.5_{0.14}$ | $61.6_{0.30}$ / $74.1_{0.24}$ | $57.8_{0.00}$ / $72.0_{0.00}$ |
| Random Matching | $53.3_{0.35}$ / $61.9_{0.15}$ | $74.2_{0.13}$ / $80.7_{0.10}$ | $51.6_{1.55}$ / $58.6_{1.57}$ | $61.5_{0.37}$ / $74.1_{0.30}$ | $57.7_{0.00}$ / $72.1_{0.00}$ |
| COMBO (ours) | $\mathbf{54.2}_{0.26}$ / $\mathbf{62.7}_{0.21}$ | $\mathbf{74.6}_{0.09}$ / $\mathbf{80.9}_{0.05}$ | $\mathbf{53.0}_{0.10}$ / $\mathbf{60.0}_{0.12}$ | $\mathbf{61.6}_{0.17}$ / $\mathbf{74.2}_{0.21}$ | $\mathbf{58.0}_{0.00}$ / $\mathbf{72.3}_{0.00}$ |

Table 6: Exact Match (EM) / F1 scores on NaturalQuestions (Kwiatkowski et al., 2019), TriviaQA (Joshi et al., 2017), WebQuestion (Berant et al., 2013), and HotpotQA (Yang et al., 2018). We conduct three runs with different random seeds and report the average and standard deviation.

that can be combined together to answer the user's question. Here's an example of your output format: Document 1: ""\n\n Document 2: ""\n\n {question}". We then parse the output to get the generated passage chains. It costs around 400 US dollars to generate 10 passage chains per question with ChatGPT for Hot-potQA.

For both the evidentiality discriminator $\mathcal{D}_{\mathcal{E}}$ and consistency discriminator $\mathcal{D}_{\mathcal{C}}$, we initialize the model from the pretrained RoBERTa-large [13] for single-hop QA and DeBERTa-large[14] for multi-hop QA from huggingface (Wolf et al., 2019). We set the max sequence length of RoBERTa as 512 and DeBERTa as 1024 since each passage chain of HotpotQA contains two passages. The training is performed on two A40 GPUs and the batch size is 16 per GPU. Peak learning rate is set to 1e-5 for RoBERTa and 5e-6 for DeBERTa. We use the Adam optimizer and linear schedule of learning rate. We train the model for 7 epochs and select the best epoch checkpoint based on the F1 score of the silver dev set, which is mined from the dev set of the original corresponding dataset. We apply sample weights in the cross-entropy loss function to handle imbalanced classes in the silver training data.

We employ FiD as the backbone architecture for our reader model in this paper. We train the models for 100k steps using four A40 GPUs with 46 GB memory and save checkpoints every 10k steps. The best checkpoint is selected based on the EM score on the dev set. For pairwise input, we set the max input length per passage pair as 400 for single-hop QA and 1000 for multi-hop QA. For single-passage input, we set the max input length per passage as 200 for single-hop QA and 500 for multi-hop QA. Per GPU batch size is 1 and we set the gradient

[13] https://huggingface.co/roberta-large
[14] https://huggingface.co/microsoft/deberta-v3-large

| Methods | NQ test | TQA test | WebQ test | HQA all $Q$ dev | HQA bridge $Q$ dev |
|---|---|---|---|---|---|
| COMBO (ours) | 54.2 | 74.6 | 53.0 | 61.6 | 58.0 |
| Same-answer matching | 55.6 | 74.8 | 53.5 | 62.4 | 58.5 |
| SOTA | $64.0^1$ | $86.1^2$ | $57.8^3$ | $68.2^4$ | - |

Table 7: Exact Match (EM) results on NaturalQuestions (Kwiatkowski et al., 2019), TriviaQA (Joshi et al., 2017), WebQuestion (Berant et al., 2013) and HotpotQA (Yang et al., 2018) for an oracle setting with access to ground-truth answers during inference. State-of-the-art (SOTA) systems — [1]ATLAS (Izacard et al., 2022), [2]PALM-2 (Anil et al., 2023), [3]UNIK-QA (Oguz et al., 2022), [4]COS (Ma et al., 2023) — leverage larger models (up to 340B) and additional knowledge.

accumulation step to 16 to imitate a large batch size. The learning rate is 1e-4 with 2000 warmup steps and a linear scheduler. Adam is the optimizer and the dropout probability is set to 0.1.

# D Additional Experimental Results and Analysis

## D.1 More Results

We also compare our proposed compatibility-guided optimal matching algorithm to an oracle setup for passage matching, in addition to the Random Matching baseline mentioned in Table 1. We first match passages into pairs if both of them contain the ground truth answer string and randomly pair the rest. We call it **same-answer matching**. Although an answer-containing passage does not necessarily contain the correct evidence (Asai et al., 2022), this setting still gives us a rough estimate of the upper bound of the model's performance. Results are shown in Table 7. We show that our method is close to the upper bound, indicating that our compatibility discriminators are good at identifying passages that contain the correct evidence.

Table 7 also includes state-of-the-art results on these datasets. It should be noted that state-of-

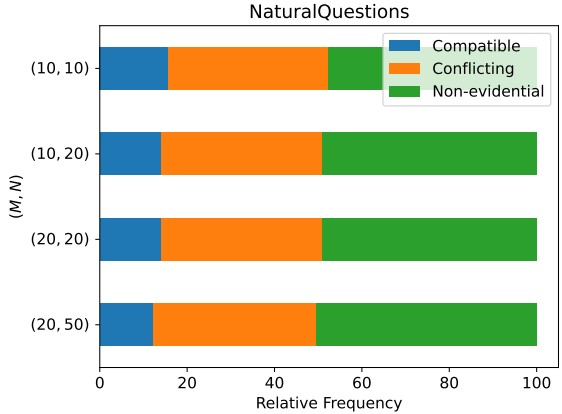

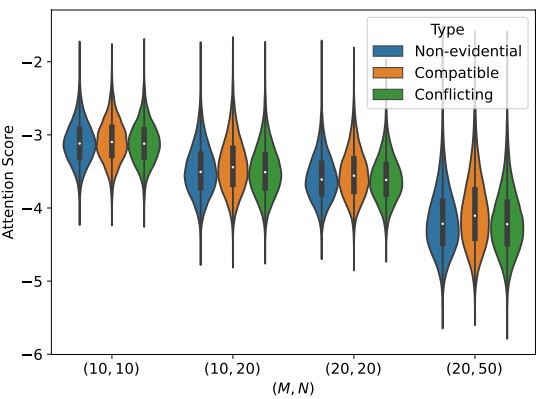

Figure 10: The relative frequency of each type of pair on the test set of NaturalQuestions w.r.t the number of LLM-generated passages ($M$) and retrieved passages ($N$) for each question. Complement to Figure 5, it shows that more knowledge conflicts arise from the increased number of input passages.

Figure 11: The attention score distribution of different types of pairs on the test set of NaturalQuestions w.r.t. the number of LLM-generated passages ($M$) and retrieved passages ($N$). Our model assigns higher attention scores to compatible pairs than incompatible ones.

the-art systems for these datasets leverage readers with much larger size (up to 340B) (Izacard et al., 2022; Anil et al., 2023) or incorporate more input retrieved passages (Ma et al., 2023) and additional knowledge in different formats (e.g., Tables, KBs) (Oguz et al., 2022). Therefore, it is not an apple-to-apple comparison between our models and theirs. However, knowledge merging can in principle benefit a wide range of open-domain QA system so our contributions should be complementary to other state-of-the-art retrievers and reader modules.

### D.2 Comparison Questions from HotpotQA are Less Sensitive to Hallucinations in LLM-generated Passages

A comparison question is one of the two types of question in HotpotQA, testing models' ability to compare two entities on some shared properties (Yang et al., 2018). This type of question is less sensitive to hallucinations in LLM-generated passages than the bridge question. For example, for the comparison question "*who is younger, Keith Bostic or Jerry Glanville?*", the ChatGPT-generated passage chain is "*Document 1: Keith Bostic was born on January 8, 1956 ... \n \n Document 2: Jerry Glanville was born on October 14, 1941 ...*". It contains hallucination errors since Keith Bostic was actually born on January 17, 1961. Nevertheless, the reader model could still make the correct prediction ("Keith Bostic") based on the fabricated evidence.

### D.3 Distribution of Pairwise Relationships

In Figure 10, we show the distribution of the relationships of passage pairs (compatible vs. conflicting vs. non-evidential) determined by our evidentiality and consistency discriminators. We observe a trend that as we increase the number of input LLM passages and retrieved passages, the percentage of compatible pairs decreases, which shows that more knowledge conflicts arise. This could account for the larger improvements of our COMBO framework over direct merging with more passages as input, which is shown in Figure 5.

### D.4 Analyzing Attention Distributions Over Different Types of Passage Pairs

In Figure 11, we show the distribution of attention scores for each type of passage pair across different numbers of input LLM-generated and retrieved passages. Following Izacard and Grave (2021a), the attention score for a passage pair is obtained by averaging the pre-normalized attention score of all tokens in the passage pair, all layers and all heads of the decoder. We find that COMBO model generally assigns higher attention scores to compatible pairs than incompatible ones. This indicates that it learns to prioritize compatible information when making its predictions, which mitigates the harmful impact of hallucination errors.