# OpenReview forum: "Merging Generated and Retrieved Knowledge for Open-Domain QA"
_EMNLP/2023/Conference — EMNLP 2023 Main_

### Official Review · Reviewer_rVZR · 2023-07-23

**Soundness:** 4

**Excitement:**

4: Strong: This paper deepens the understanding of some phenomenon or lowers the barriers to an existing research direction.

**Paper Topic And Main Contributions:**

The paper introduces a compatibility-oriented knowledge merging framework (COMBO) that addresses the challenge of merging retrieved and LLM-generated knowledge in open-domain QA [1]. Unlike previous methods that tend to favor generated passages over retrieved ones when conflicts arise, COMBO combines both types of information to promote the usage of factual and relevant content. The authors propose 'compatible scores' based on consistency discriminator and evidentiality discriminator, enabling the identification of compatible pairs of retrieved and generated passages. The experimental results demonstrate that the proposed approach outperforms the direct merging baseline, especially in scenarios with a high conflicting rate.

[1] Generate rather than Retrieve: Large Language Models are Strong Context Generators (Yu et al., ICLR'23)

**Reasons To Accept:**

1. The paper is well-written and presents the concepts in a clear and accessible manner. The use of visually appealing illustrations aids in understanding the proposed framework.
2. The experimental evaluation shows that the proposed COMBO method achieves consistently strong results across various datasets, including both single-hop and multi-hop scenarios.
3. Through detailed analysis, the paper demonstrates that the COMBO method effectively addresses conflicts between retrieved and generated passages, leading to improved performance compared to direct merging.



**Reasons To Reject:**

While the COMBO method exhibits significant improvements over the direct merging baseline, the paper could further discuss the performance gain compared to the "Random Matching" method. Clarifying the margin of improvement in this aspect would provide a more comprehensive perspective on the method's efficacy.

**Reproducibility:**

5: Could easily reproduce the results.

**Reviewer Confidence:**

4: Quite sure. I tried to check the important points carefully. It's unlikely, though conceivable, that I missed something that should affect my ratings.

---

> ### Author Rebuttal · Authors · 2023-08-25
>
> Thank you for the helpful comments. Please see our responses below.
>
> ***Re: The paper could further discuss the performance gain compared to the "Random Matching" method***
>
> Please see our response under “***Re: Incremental results***” to Reviewer KaNa.

---

### Official Review · Reviewer_mizo · 2023-08-05

**Soundness:** 4

**Excitement:**

3: Ambivalent: It has merits (e.g., it reports state-of-the-art results, the idea is nice), but there are key weaknesses (e.g., it describes incremental work), and it can significantly benefit from another round of revision. However, I won't object to accepting it if my co-reviewers champion it.

**Paper Topic And Main Contributions:**

This paper proposes a new method named COMBO for merging content from retrieved and LLM-generated passages for open-domain QA. A primary goal of the work is to leverage the parametric knowledge of LLMs while making sure they do not contradict with the content of retrieved passages. COMBO uses two discriminators to identify “compatible” pairs of retrieved and LLM-generated passages, and then feeds them to a fusion-in-decoder (FiD) reader to produce the final answer. The authors also present approaches to construct silver training data for the two discriminators. Evaluated on a number of test sets, COMBO outperforms various relevant baselines, including other methods for merging retrieved and LLM-generated content. The authors also present several ablation and qualitative results.


**Questions For The Authors:**

1. In lines 490—494, did you try the fixing the order to (rp_j, lp_i)?
2. How did you generate different passages for the same question using LLMs? Include the details of sampling in the paper.
3. (This is a suggestion, not a question:) F1 score is a more stable and reliable QA metric than EM. Consider also reporting F1 scores.

**Reasons To Accept:**

1. The paper proposes a novel and interesting approach to merging retrieved and LLM-generated content for downstream QA.
2. The experimental protocol is sound and thorough.
3. The paper is very clearly written — I enjoyed reading it.

**Reasons To Reject:**

The performance gain over random matching is quite small (and also apparently not statistically significant). Given the considerable amount of additional effort COMBO requires and also its higher parameter count (due to the two discriminators), such small improvements in quality make me question the practicality of the approach.

**Reproducibility:**

4: Could mostly reproduce the results, but there may be some variation because of sample variance or minor variations in their interpretation of the protocol or method.

**Reviewer Confidence:**

4: Quite sure. I tried to check the important points carefully. It's unlikely, though conceivable, that I missed something that should affect my ratings.

**Typos Grammar Style And Presentation Improvements:**

Lines 196—197: “all possible passage pairs” -> “all possible (retrieved, LLM-generated) passage pairs”

---

> ### Author Rebuttal · Authors · 2023-08-25
>
> Thank you for the helpful comments. Please see our responses below.
>
> ***Re: The performance gain over random matching is quite small***
>
> Please see our response under “***Re: Incremental results***” to Reviewer KaNa.
>
> ***Re: Given the considerable amount of additional effort COMBO requires and also its higher parameter count (due to the two discriminators), such small improvements in quality make me question the practicality of the approach***
>
> Please see our response under “***Re: Requires extra training***” to Reviewer KaNa.
>
>
> ***Re: Question 1***
>
> We do not experiment with flipping the order of generated and retrieved passages. However, given the total input length is relatively short (less than 400 tokens per pair), we would expect similar results as long as the relative order is fixed.
>
>
> ***Re: Question 2***
>
> We use the same prompt and generate passages by multiple decoding passes with nucleus sampling. We will include detailed hyperparameters of the sampling process in camera-ready. We also refer our readers to the GenRead (Yu et al., 2023) paper for details since we use their provided passages directly.
>
>
> ***Re: Question 3***
>
> Per your suggestion, we report the F1 metric as follows. Basically, the trend is similar to EM results.
>
> **Table II**: F1 results on NaturalQuestions, TriviaQA, and WebQuestion.
> | Methods    | NQ | TQA | WebQ |
> | -------- | ------- | ------- | ------- |
> | Direct Merging  |  61.2±0.27  |  80.5±0.11 |  58.5±0.11 |
> | Random Matching |   61.9±0.12   |  80.7±0.08 |  58.6±1.28 |
> | COMBO    |  62.7±0.17  | 80.9±0.04  | 59.3±0.48 |
>
>
> **References**
>
> [1] Wenhao Yu, Dan Iter, Shuohang Wang, Yichong Xu, Mingxuan Ju, Soumya Sanyal, Chenguang Zhu, Michael Zeng, and Meng Jiang. 2023. Generate rather than retrieve: Large language models are strong context generators. 11th International Conference on Learning Representations. 2023

---

### Official Review · Reviewer_KaNa · 2023-08-05

**Soundness:** 5

**Excitement:**

3: Ambivalent: It has merits (e.g., it reports state-of-the-art results, the idea is nice), but there are key weaknesses (e.g., it describes incremental work), and it can significantly benefit from another round of revision. However, I won't object to accepting it if my co-reviewers champion it.

**Paper Topic And Main Contributions:**

The paper proposed a compatibility-based approach for open-domain QA that merges both retrieved and generated knowledge. A framework that follows previous work (Asai et al., 2022) for mining silver labels is used to train a compatibility scorer, which contains two components, evidentiality and consistency. The experiments show that the proposed method improves upon simply merging retrieved and generated knowledge, and extensive ablation studies reveal the usefulness of each component.

**Questions For The Authors:**

Question A: Are the discriminators transferable across different dataset?
Question B: Why not include results of filtering pairs with zero evidentiality?
Question C: Is the improvement of COMBO over “Random Matching” statistically significant?
Question D: The definition of conflict rate is a bit confusing. Passages not containing the answer are not necessarily contradicting the gold passages; they might just not include the information needed for answering the question.
Question E: What’s the percentage of predictions that are sub-spans of at least one of the retrieved passages? Some predictions could potentially be sub-spans of both generated and retrieved passages, so 65% does not necessarily mean the reader favors generated passages.
Question F: The authors describe a subset of examples where the initial predictions are correct using Retrieved-passage-only method, but fooled by adding LLM-generated passages. I wonder how much of these fooled examples could be un-fooled using the proposed method? (Meaning “Retrieved-passage-only” is correct, “Direct Merging” is incorrect, and “COMBO” is correct again).


**Reasons To Accept:**

1. The idea is sound: The mathematical derivation of compatibility, evidentiality and consistency is clear and sound. The intuition behind each design is carefully described.
2. Solid experiments: The authors provided meaningful baselines to the experiments, and report the SOTA results in the appendix. The results come with statistical significance. There are extensive ablation studies and results of using different numbers of passages. The conflicting rate experiment (Table 3) also provides a possible explanation of the performance improvement.


**Reasons To Reject:**

There are three points in total, ranked from major to minor:

1. Incremental results: The performance improvement over “Direct Merging” is already incremental, and yet the boost over “Random Matching” is even smaller. This indicates that most improvement comes from matching retrieved and LLM-generated passages. Also, Table 6 shows the upper bound is not too far from the proposed approach, meaning little room for improvement. The bottleneck might be generating better related passages.
2. Requires extra training: Both the evidentiality and consistency discriminator are separately trained on each dataset. This implies that an extra training step is required each time the model is tested on a new dataset. Given the limited performance gain, it is difficult to justify the training cost.
3. Discriminators might not work out-of-domain: There is no gold label for the discriminators, and thus it is difficult to quantify their performances. However, it is conceivable that using the discriminator trained on NQ might not transfer to TQA, for example. This limits the use case of the framework, and it may not work with datasets with fewer silver labels, as mentioned in the limitation section.


**Reproducibility:**

4: Could mostly reproduce the results, but there may be some variation because of sample variance or minor variations in their interpretation of the protocol or method.

**Reviewer Confidence:**

4: Quite sure. I tried to check the important points carefully. It's unlikely, though conceivable, that I missed something that should affect my ratings.

---

> ### Author Rebuttal · Authors · 2023-08-25
>
> Thank you for the helpful comments. Please see our responses below.
>
> ***Re: Incremental results***
>
> The experiment results in Table 1 of the submitted paper are based on a single run of models. Considering the potential for random seeds to influence these outcomes, to the best of our efforts, we try to mitigate the uncertainty by repeating each experiment for 3 runs. Here, we offer the average and standard deviation (std.) of Exact Match (EM) metrics, over three single-hop QA datasets.
>
> **Table I**: Exact Match results on NaturalQuestions, TriviaQA, and WebQuestion.
> | Methods    | NQ | TQA | WebQ |
> | -------- | ------- | ------- | ------- |
> | Direct Merging  |  52.4±0.80  |  74.2±0.08 |  51.1±0.30 |
> | Random Matching |   53.3±0.32   |  74.3±0.21 |  51.6±1.55 |
> | COMBO    |  54.2±0.20  | 74.6±0.07  | 52.3±0.60 |
>
> We observe a similar trend that our proposed method, COMBO, significantly outperforms both baselines. The improvement of Random Matching over Direct Merging accentuates the value of pairwise input formulation in aiding the reader model to encode knowledge from dual sources. Then we take a further step to enhance performance by composing compatible pairs.
>
> We also note that any potential performance improvements are somewhat constrained by the percentage of the hallucinated (generated) and evidential (retrieved) passages provided. In the extreme scenario where all generated passages are devoid of hallucinations or all retrieved passages lack correct evidence, our method would default to Random Matching, as the compatibility matching score would be uniform.
>
> In the conditions of our experiment, we did not manipulate the distribution of questions or input passages in any way. However, we anticipate that in more controlled settings, such as PopQA (Mallen et al., 2023) where questions about infrequent entities are examined in more detail, our method could yield more dramatic performance improvements. At present, LLMs appear more susceptible to hallucinating knowledge pertaining to less frequent entities. We eagerly anticipate investigating this aspect further in our future work.
>
>
> ***Re: Requires extra training***
>
> To reduce the effort of training a separate discriminator for each dataset, we additionally investigate the performance of a unified discriminator. Specifically, we train a discriminator on silver labels mined from both NQ and TQA and report its results on NQ and TQA separately (row ‘COMBO’ in Table I). We also demonstrate a use case for smaller datasets where fewer labels could be mined. For results on WebQ of Table I, we further finetune the unified discriminator with WebQ’s data. The performance gains over baselines show the effectiveness of training a generalizable discriminator that works for diverse datasets.
>
> Regarding the efficiency of our pipeline, we argue that discriminator training essentially requires one initial setup prior to its deployment across various datasets. The discriminator model can be any encoder-only model in principle and easily fits into most GPUs available in school labs. We would also make our checkpoints publicly accessible so future NLP practitioners don’t have to train from scratch again.
>
> One future direction is to employ an LLM with in-context learning to compute the compatibility scores instead of training a discriminator from scratch, thereby eliminating the necessity for (weakly) supervised training. This makes our approach generalizable to a few-shot setting. But it would also require carefully examining whether and which LLM is able to discern the knowledge conflicts between its own parametric memory and retrieved evidence.
>
>
> ***Re: Discriminators might not work out-of-domain***
>
> In “***Re: Requires extra training***”, we demonstrate our discriminators’ generalizability under in-domain setups for datasets of diverse domains. We will add the results of discriminators trained on dataset X and tested on dataset Y in camera-ready. We do anticipate that the out-of-domain performance will not be as good as the in-domain one shown in Table I.
>
> Regarding your concern that “*it is difficult to quantify the discriminators’ performances*”, we encourage you to check out Appendix D.4, where we conduct a human evaluation of the discriminators’ performance. We find that our discriminators yield an overall accuracy of 78% on their predicted labels. It shows that our discriminators are capable of providing signals of passage compatibility to the reader model for making well-informed decisions.
>
> In Table I, we show that “*for datasets with fewer silver labels*”, e.g. WebQ, we can warmup the discriminator training with silver labels mined from other similar datasets.
>
> ***Re: Question A***
>
> We plan to incorporate the results of the transferred discriminator (trained on NQ and tested on TQA) in camera-ready in case of acceptance. Please also see “***Re: Requires extra training***” for a unified discriminator experiment.
>
>
> ***Re: Question B***
>
> In the early stage of this project, we experimented with some simple strategies for filtering passages based on the evidentiality and consistency scores, but they did not work well empirically due to the noise of discriminators trained on silver labels. For simplicity, we include all passages in this project, leaving it to future work to filter passage pairs. We note that previous work (Asai et al., 2022) also does not filter non-evidential passages but instead includes all of them during prediction.
>
>
> ***Re: Question C***
>
> For the paper results based on a single run of the model, the improvement of COMBO over “Random Matching” is not statistically significant. However, randomness of model training may play a role here so we encourage you to check out Table I for more reliable results based on 3 runs. We highlight that the average performance of COMBO is still higher than Random Matching.
>
>
> ***Re: Question D***
>
> We would like to clarify the assumptions behind the definition of the conflicting rate. According to Assumption 2 (lines 215-217) and our empirical observations in Appendix A.4, the LLM-generated passage almost always contains a plausible answer to the question, be it factual or hallucinated. Based on this assumption, LLM passages not containing the gold answer ($M-M_{A}$) are likely to contain a hallucinated answer, which contradicts the evidential retrieved passages. We believe you try to argue this point for the *retrieved* passages, which “*may not include the information needed for answering the question*”. Therefore, we still think our definition is conceptually valid. However, our assumptions may not always hold true so we agree that the conflicting rate is only a proxy for the actual situation.
>
>
> ***Re: Question E***
>
> 26% of predictions are sub-spans of at least one of the retrieved passages on NQ. This indicates a certain level of resilience to knowledge conflicts within the base reader model, however, it could be further improved through our proposed approach.
>
>
> ***Re: Question F***
>
> Our proposed method successfully corrects the answers on 41.0% of the examples that fool the model.
>
>
> **References**
>
> [1] Mallen, Alex, et al. "When not to trust language models: Investigating effectiveness of parametric and non-parametric memories." *Proceedings of the 61st Annual Meeting of the Association for Computational Linguistics (Volume 1: Long Papers)*. 2023.
>
> [2] Asai, Akari, Matt Gardner, and Hannaneh Hajishirzi. "Evidentiality-guided Generation for Knowledge-Intensive NLP Tasks." *Proceedings of the 2022 Conference of the North American Chapter of the Association for Computational Linguistics: Human Language Technologies*. 2022.

---

### Meta-Review · Area_Chair_5JRB · 2023-09-15

**Recommendation:** 4

**Metareview:**

This paper presents an approach for combining LM generated and retrieved support passages in open-domain QA to balance coverage and faithfulness. The paper solves an important problem, and the reviewers all agreed that the approach is sound and that the paper is well-written. The reviewers were moderately excited about this work because the improvements over baselines were small particularly given that the approach requires additional training.

I think this is solid work, and while the improvements are small, it does provide an interesting solution for the coverage-faithfulness tradeoff by combining retrieval and generation.

---

### Decision · Program_Chairs · 2023-10-07

**Decision:**

Accept-Main

**Comment:**

This paper presents an approach for combining LM generated and retrieved support passages in open-domain QA to balance coverage and faithfulness. The paper solves an important problem, and the reviewers all agreed that the approach is sound and that the paper is well-written. The reviewers were moderately excited about this work because the improvements over baselines were small particularly given that the approach requires additional training.

I think this is solid work, and while the improvements are small, it does provide an interesting solution for the coverage-faithfulness tradeoff by combining retrieval and generation.